# Peripheral nerve regeneration following scaffold-free conduit transplant of autologous dermal fibroblasts: a non-randomised safety and feasibility trial

Ryosuke Ikeguchi [1,2 ✉], Tomoki Aoyama[3], Takashi Noguchi[2], Mika Ushimaru[4], Yoko Amino[4], Akiyoshi Nakakura[4], Noriko Matsuyama[4], Shiori Yoshida[4], Momoko Nagai-Tanima [3], Keiko Matsui[5], Yasuyuki Arai [5], Yoko Torii[6], Yudai Miyazaki[6], Shizuka Akieda[6] & Shuichi Matsuda[2]

## Abstract

**Background** The use of Bio 3D nerve conduits is a promising approach for peripheral nerve reconstruction. This study aimed to assess their safety in three patients with peripheral nerve defects in their hands.

**Methods** We describe a single institution, non-blinded, non-randomised control trial conducted at Kyoto University Hospital. Eligibility criteria included severed peripheral nerve injuries or a defect in the region distal to the wrist joint not caused by a congenital anomaly; a defect with a length of ≤20 mm in a nerve with a diameter ≤2 mm; failed results of sensory functional tests; ability to register in the protocol within 6 months from the day of injury; refusal of artificial nerve or autologous nerve transplantation; age 20–60 years; and willingness to participate and provide informed written consent. Six weeks before transplantation, skin was harvested, dermal fibroblasts were isolated and expanded, and Bio 3D nerve conduits were created using a Bio 3D printer. Bio 3D nerve conduits were transplanted into the patients' nerve defects. The safety of Bio 3D nerve conduits in patients with a peripheral nerve injury in the distal part of the wrist joint were assessed over a 48-week period after transplantation.

**Results** No adverse events related to the use of Bio 3D nerve conduits were observed in any patient, and all three patients completed the trial.

**Conclusions** Bio 3D nerve conduits were successfully used for clinical nerve reconstruction without adverse events and are a possible treatment option for peripheral nerve injuries.

## Plain language summary

Physical injuries often result in damage to nerves, for example, in the hands. Replacement of the nerve with nerves removed from elsewhere in the patient's body is often the suggested treatment when the nerve is unable to repair itself. As an alternative to remove healthy nerve from elsewhere in the body, we used an adapted printer to create an artificial nerve equivalent from skin cells obtained from the patient's skin. We reconstructed the nerves of three individual with nerve defects in their hands, and we found that the function of the nerve improved, and the people did not experience negative consequences. This approach could be used widely to repair damaged nerves.

[1] Department of Rehabilitation Medicine, Kyoto University Graduate School of Medicine, Kyoto, Japan. [2] Department of Orthopaedic Surgery, Kyoto University Graduate School of Medicine, Kyoto, Japan. [3] Human Health Sciences, Graduate School of Medicine, Kyoto University, Kyoto, Japan. [4] Institute for Advancement of Clinical and Translational Science, Kyoto University Hospital, Kyoto, Japan. [5] Center for Research and Application of Cellular Therapy, Kyoto University Hospital, Kyoto, Japan. [6] Cyfuse Biomedical K.K, Tokyo, Japan. ✉email: ikeguchi@kuhp.kyoto-u.ac.jp

Previous studies have reported the creation of Bio 3D nerve conduits using a Bio 3D printer (Cystrix®, Cyfuse Biomedical KK, Tokyo, Japan) to promote peripheral nerve regeneration in experimental models[1–7]. Bio 3D printing is a novel technology that shapes cell spheroids three-dimensionally by placing them onto thin needles according to pre-designed three-dimensional data[8]. This fabricated three-dimensional tissue comprises only cells with no artificial scaffolds. Blood vessels, cartilage, bone, and trachea have been fabricated using this technology[1,8–10]. However, Bio 3D nerve conduits have not been used for peripheral nerve defect treatment in clinical settings. To the best of our knowledge, in this study, for the first time, we apply Bio 3D nerve conduits in human peripheral nerve reconstruction.

Peripheral nerve injuries are common traumatic injuries of the upper and lower extremities. Autologous nerve grafting is the gold standard for bridging nerve gaps when direct nerve repair is not possible because of traumatic nerve defects[11,12]. However, autologous nerve grafts need healthy nerve harvesting, which may lead to donor site morbidity, such as sensory loss and painful neuroma formation[1,2]. As an alternative, Bio 3D nerve conduits were developed using a Bio 3D printer in experimental models[1–7]. Bio 3D nerve conduits have been shown to bridge peripheral nerve defects because their cellular elements promote better peripheral nerve regeneration than artificial nerve conduits[7]. Additionally, the proof-of-concept study and long-term safety of Bio 3D nerve conduits have been reported[4,6].

Accordingly, we conducted an investigator-initiated clinical trial and reconstructed peripheral nerve defects in the hands using Bio 3D nerve conduits. This investigation confirmed the safety and good nerve regenerative ability of the Bio 3D nerve conduit.

## Methods

**Study setting**. This was a single institution, non-blinded, non-randomised control trial. Safety and efficacy of Bio 3D nerve conduits in patients with a peripheral nerve injury in the distal part of the wrist joint were assessed by measuring motor and sensory nerve function in the affected upper limb over a 48-week period after transplantation. The trial was conducted at Kyoto University Hospital, and 3D nerve conduits were produced in the cell processing centre at the Center for Cell and Molecular Therapy (CCMT) at Kyoto University Hospital.

**Ethics statement**. This study was conducted in accordance with the study protocol and adhered to the principles of the Declaration of Helsinki and Good Clinical Practice (GCP) guidelines. This study was approved by the Institutional Review Board of Kyoto University Hospital on 25 March 2020 (K069, protocol version 1.0, February 2020) in agreement with the Pharmaceuticals and Medical Devices Agency, Japan. The protocol was registered with the Japan Registry of Clinical Trials (jRCT2053200022, Registered on 1 June 2020, https://jrct.niph.go.jp/en-latest-detail/jRCT2053200022). The investigator obtained informed consent from all participants housed at Kyoto University Hospital. All participants provided written informed consent to participate in the study (Fig. 1a). Our first patient was admitted on 22 December 2020 and the last patient on 4 October 2021.

**Inclusion and exclusion criteria**. Patients fulfilling the following criteria were included: (1) severed peripheral nerve injuries or a defect in the region distal to the wrist joint not caused by a congenital anomaly; (2) a defect with a length of ≤20 mm in a nerve with a diameter ≤2 mm; (3) failed results of sensory functional tests, including the Semmes–Weinstein monofilament test (SWT)[13] and static and moving 2-point discrimination sensory functional tests (s2PD and m2PD, respectively)[14], for dermatome distribution of the injured peripheral nerve; (4) able to register in the protocol within 6 months from the day of injury; (5) refused artificial nerve or autologous nerve transplantation; (6) age 20–60 years; and (7) willingness to participate and provide informed written consent (Fig. 1b). All trial participants consented to the publication of any identifying information.

Patients fulfilling any of the following criteria were excluded: (1) peripheral nerve injury, including those in the fingers affected by infection and severe damage of accessories—including the skin, tendon, and bone—injury at multiple sites of the nerve and wide area, and direct suture-feasible; (2) presence of antibodies against hepatitis B, human immunodeficiency, or human T-cell leukaemia virus; (3) active infection, such as hepatitis C, syphilis (*Treponema pallidum* antibody-positive in serological tests for syphilis), and human parvovirus B19; (4) a history of allergy or anaphylaxis reaction to a component(s) of the clinical trial products, such as aminoglycoside antibiotics, polyene macrolide antibiotics, bovine serum, and/or metal(s); (5) one of the following complications, including cardiovascular disease(s), diabetes mellitus, stroke (including history), cervical spondylosis, cervical myelopathy, polyneuropathy, Guillain–Barre syndrome, amyotrophic lateral sclerosis, peripheral circulatory failure, rheumatoid arthritis, collagen disease, depression, schizophrenia, automatic neuropathy, or dementia; (6) malignant disease and/or medical history thereof; (7) previous treatment with immunosuppressive agents and/or steroids excluding local effects; (8) simultaneous participation in another interventional trial and/or a clinical trial within the previous 3 months before enrolment in this trial; (9) history of participation in studies investigating the transplant of clinical trial products; (10) pregnant females, those lactating, and those unwilling to prevent pregnancy during the study period; and (11) individuals judged by the attending physician to be unfit or not suitable for the study.

**Bio 3D nerve conduit manufacturing**. Skin tissue (>1 cm²) from each patient was biopsied from the abdomen or inguinal region (Fig. 2a) under local anaesthesia and transferred to a cell processing centre. Skin tissues were washed with phosphate-buffered saline (PBS) containing amphotericin B (019-13364, Fujifilm corp. Tokyo, Japan), and the subcutaneous fat layers were thoroughly resected. Treated skin tissues were immersed in DMEM low glucose medium (044-33555, Fujifilm) containing fibroblast growth supplements (CC-3132, LONZA, Walkersville, MD, USA), foetal bovine serum (FBS, 12007C-500ML, Nichirei Biosciences Inc.), and antibiotics containing 1000 units mL⁻¹ of dispase II (9001-92-7, Fujifilm) and incubated at 37 °C for 2–3 h. The epidermal layers of the skin sections were removed, and the remaining tissue was chopped into small pieces, measuring 1–2 mm or less, and incubated in a humidified 5% $CO_2$ and 37 °C incubator. The culture medium (DMEM low glucose medium, FBS, fibroblast growth supplements, and antibiotics) was replaced every 2–3 days. The dermal fibroblasts from passages 4–5 were used in this study (Fig. 2b). Cell surface adhesion molecule (CD90) and haematopoietic marker (CD45) were used as negative controls and assessed using flow cytometry. Cells detached using trypsin EDTA (203-20251, Fujifilm) were washed in PBS containing FBS and incubated with anti-CD90 (562556, BD Biosciences) and anti-CD45 (560178, BD Biosciences) antibodies for 30 min on ice. After staining, cells were acquired on BD FACS Canto™ II (BD Biosciences) and analysed using BD FACSDiva (BD Biosciences).

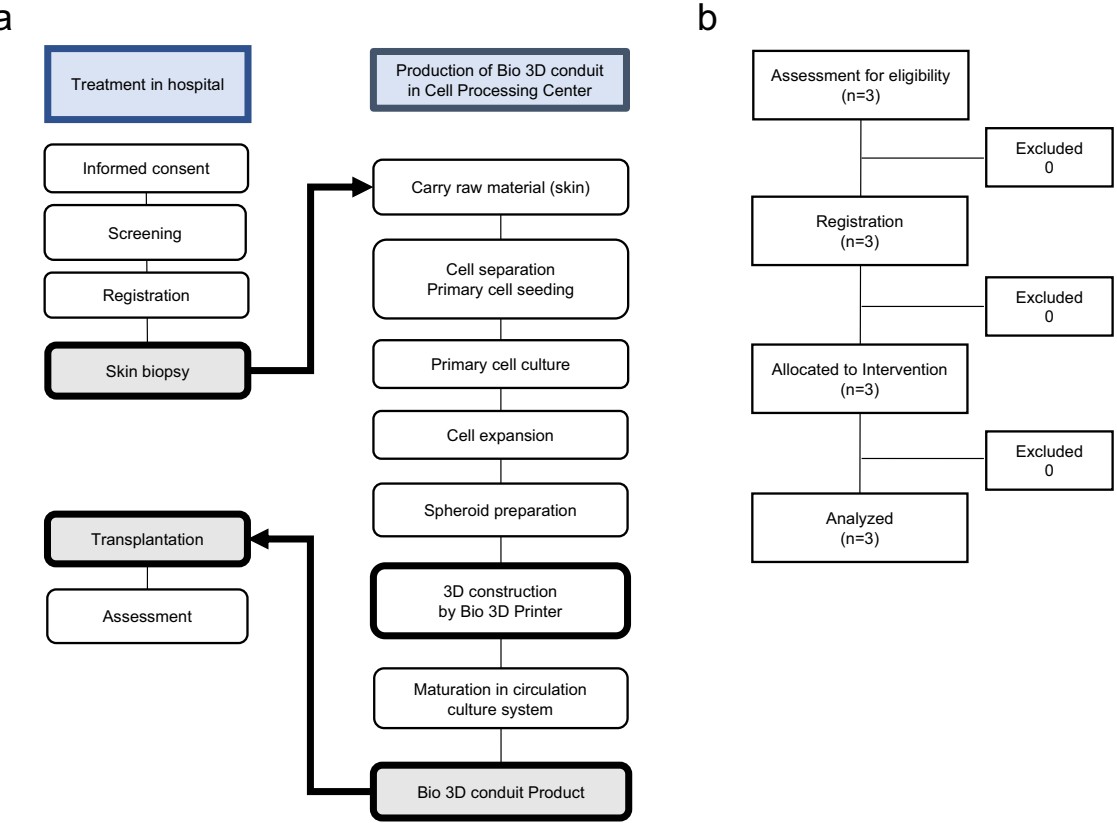

**Fig. 1 Study design and Bio 3D nerve conduit creation. a** Flow diagram of treatments and Bio 3D nerve conduit creation. **b** CONSORT flow diagram.

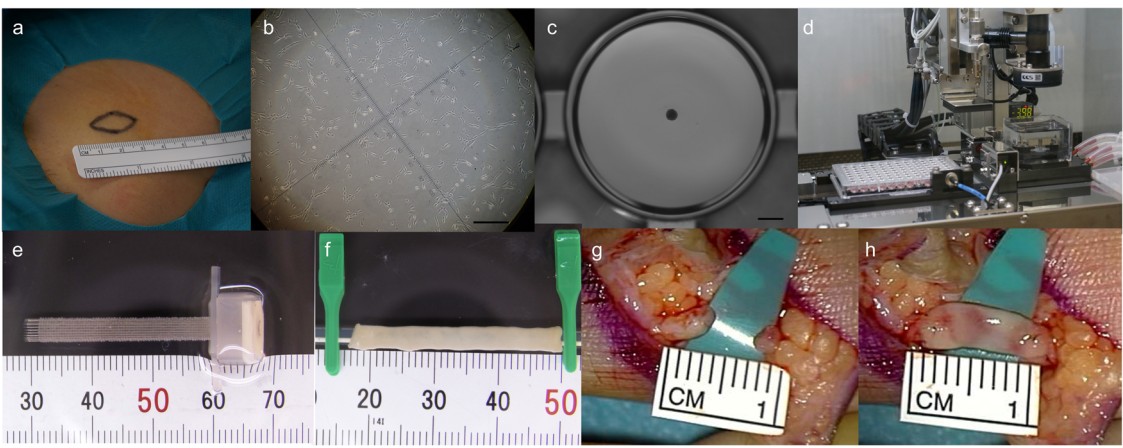

**Fig. 2 Bio 3D nerve conduit manufacturing and transplantation. a** Bio 3D nerve conduit creation. **b** Skin tissue biopsied from the inguinal region.
**c** Expanded dermal fibroblasts. Scale bar: 10 mm. **d** Aggregation of homogeneous multicellular spheroids. Scale bar: 1000 μm. **e** Assembly of the conduits by robotically aspirating spheroids into a fine suction nozzle from the 96-well plate using Bio-3D Printer. **f** Tubular structure according to the pre-designed 3D model. **f** The mature conduit structure created measures >20 mm in length and 2.5 ± 1 mm in diameter. **g** The nerve defect. **h** The nerve gap bridged by 3D conduit.

The conduits were fabricated from dermal fibroblasts using a Bio-3D Printer (Cystrix®, Cyfuse Biomedical KK)[3–9]. Cells were detached and collected using trypsin EDTA, centrifuged, and resuspended in a minimal volume of fresh media. An appropriate amount of the cell suspension, at a concentration of approximately $1 \times 10^5$ cells mL$^{-1}$, was incubated in a low-cell-adhesion 96-well plate (SUMILON PrimeSurface® MS-9096U, Sumitomo Bakelite). After 48 h, cells aggregated to form homogeneous multicellular spheroids with diameters of $500 \pm 50$ μm (Fig. 2c). The conduits were assembled by robotically aspirating spheroids

into a fine suction nozzle from the 96-well plate. Skewers of a circular needle array (NA 1027, Cyfuse Biomedical KK) and Bio-3D Printer (Cystrix®, Cyfuse Biomedical KK) (Fig. 2d) were used to develop a tubular structure according to a pre-designed 3D model (Fig. 2e).

One week after the procedure, adjacent spheroids were conglutinated to create a Bio 3D conduit, and the needle array was removed. The obtained Bio 3D conduits were transferred to a perfusion bioreactor, in which a silicon tube with an external diameter of 2.5 mm was placed inside each Bio 3D conduit.

Perfusion cultivation was continued until the desired function and strength were achieved. The mature conduit structure was >20 mm in length and 2.5 ± 1 mm in diameter (Fig. 2f).

**Transplantation of the Bio 3D nerve conduit**. Under general anaesthesia, the injured nerve was exposed by removing the subcutaneous scar tissue and creating an operational area. Under an operative microscope, the transected nerve was dissected, scarred nerve segments were resected from both proximal and distal nerves to freshen the nerve stump, and the nerve gap was measured (Fig. 2g). Three surgeries were performed by two microsurgery specialists. Each patient's 20-mm Bio 3D conduit was cut to adjust the conduit length for interposition between the proximal and distal stumps. Each stump was then pulled 1.5 mm into the conduit and anchored in place using epineural 10–0 nylon sutures, bridging the inter-stump gap in the conduit (Fig. 2h)[3–7]. After transplantation, the operation site was closed with disinfection, and immobilisation was performed for 3 weeks.

**Primary endpoints**. Safety assessments were scheduled at 1, 4, 8, 12, 16, 20, 24, 28, 32, 36, 40, 44, and 48 weeks after transplantation to address the primary objective (Supplementary Table 1). The participants were interviewed about their condition, and vital signs, including blood pressure, pulse, and body temperature, were measured. Biochemical tests were performed at 4, 12, 24, 36, and 48 weeks after transplantation to monitor the condition of the affected forelimb and the body response. The occurrence of adverse events and information regarding the combination treatment was confirmed. All events and clinical dates were recorded and analysed according to the Medical Dictionary for Regulatory Activities (MedDRA) version 25.1, including the date of the initial event, policy decision date, details of adverse event(s), the total number of adverse events, and the total number of follow-up days (number of days in the observation period). The data centre was informed of all adverse events, such as the number of patients who experienced at least one adverse event and those who discontinued treatment due to adverse events, and regulatory requirements were followed for handling them. The safety assessment committee investigated cases where the endpoints may have been affected.

**Secondary endpoints**. In addition to the safety endpoints, exploratory endpoint assessments were performed to evaluate the efficacy of Bio 3D nerve conduit at 4, 12, 24, 36, and 48 weeks after transplantation.

**Semmes–Weinstein Monofilament Test (SWMT)**[13,15]. Twenty Semmes–Weinstein monofilaments were prepared from nylon 612 (SOT-DM20A, SAKAI Medical Co. Tokyo, Japan) and used for sensory evaluation[13,15]. The SWMT measures the response to the touching sensation of monofilaments according to a numerical quantity. The filament tip was placed 2 cm away from the skin and vertically down on the skin for 1.5 s. The filament tip contacted the skin, and a force was applied to bend the filament at a right angle. The force was then removed to straighten the filament while keeping the tip in contact with the skin. The filament tip was placed away from the skin for 1.5 s. Based on these results, grading was performed into excellent, good, and poor categories[13].

**Static 2-point discrimination (s2PD) and movement 2-point discrimination (m2PD) test**[14]. The s2PD test was performed using the Dellon–Mackinnon discriminator tool (B003TMB168, Fabrication Enterprises Inc.). The pointers were adjusted such that the centre of the tips varied from 1 to 25 mm apart. The final

threshold value for each digit was determined as the smallest distance at which the patient could correctly discriminate between one and two points[14].

Furthermore, the m2PD test was performed using the Dellon–Mackinnon Discriminatory tool. The pointers were moved along the surface of the finger from the proximal to the distal positions. The final threshold value was determined as s2PD. For each test, grading was performed into excellent, good, or poor based on the results[14].

**Reference endpoints**
*Shortened disabilities of the arm, shoulder, and hand questionnaire (QuickDASH)*[16,17]. The QuickDASH questionnaire is an 11-item questionnaire used to assess physical function and symptoms in individuals with any or multiple upper limb musculoskeletal disorders[16,17]. Higher values correspond to greater disability/severity of symptoms.

*Motor nerve function analysis*. Motor nerve function of the peripheral nerve of the upper limb was evaluated using the assessment of the Perfect O sign, Froment's sign, and manual muscle test (MMT), commonly used in clinical practice[18]. The collapse of the Perfect O sign[19] reflects reduced functionality and weakness in the muscles innervated by the median nerve. Froment's sign[20] is a physical examination of the hand that tests for palsy or ulnar nerve disability. It evaluates the strength of the adductor pollicis of the thumb, which is innervated by the ulnar nerve. A positive sign indicates reduced functionality and muscle weakness in the pinch grip[20]. The MMT was used as a motor nerve functional test to quantitatively assess the maximum force of the muscles associated with the median and ulnar nerves. The MMT is widely used to evaluate the maximum force a muscle can generate, and its reliability has been confirmed[18]. It is scored on a scale from 0 to 5, with grade 5 indicating normal muscle function and grade 0 indicating complete paralysis[21].

**Data collection and analysis**. The investigators completed the case report forms in a booklet while following the instructions. Each completed booklet was submitted to the data centre. The data in the booklet were entered into a database using the double-entry method. The data quality was validated by checking for missing and out-of-range values.

The safety analysis group included participants who underwent a skin biopsy after registration. The number of adverse and faulty events related to the products or processes of transplantation after skin biopsy were counted according to MedDRA J ver. 25.1, and the percentage of each event was calculated using the results from all samples. An independent data monitoring committee was established to assess safety data if serious adverse events occurred and evaluate whether the per-protocol set required any modification. A qualified and independent auditor was appointed to audit the trial systems and conduct trials before and during the study per a written procedure.

The results of the sensory functional analysis were used to evaluate efficacy. All analyses, including the trend of changes for each participant at each time point, were performed using SAS version 9.4 (SAS Institute, Cary, NC, USA). Individuals who committed a serious violation of the study process or GCP and/or demonstrated non-compliance after registration were excluded.

**Reporting summary**. Further information on research design is available in the Nature Portfolio Reporting Summary linked to this article.

## Results and discussion

In this study, we included three patients aged ≥20 years with peripheral nerve defects in their hands. The patients included a 46-year-old male with a severe defect in the superficial branch of his left ulnar nerve, a 50-year-old male with a digital nerve defect in his left hand's ring finger, and a 34-year-old male with a nerve defect in the superficial branch of his left dorsal hand's radial nerve (Table 1). Six weeks before transplantation, 1 ×1-cm$^2$ skin was harvested, dermal fibroblasts were isolated and expanded, and Bio 3D nerve conduits were created using a Bio 3D printer. Perioperatively, the lacerated nerves were exposed to measure the nerve defect size, and Bio 3D nerve conduits were transplanted to bridge the nerve defects. Adverse events and nerve recovery were assessed during follow-up.

No adverse events related to the 3D conduits were observed in any patient from skin harvesting until 48 weeks post-transplantation. The only adverse events included surgical wound pain (Table 2). No patient showed signs of infection or allergy at the surgical site. All patients reported sensory improvement during postoperative follow-up. Figure 3 shows the patients' sensory recovery (Fig. 3a, c, d). All patients had poor pre-surgery Semmes–Wein monofilament test scores (6.65). However, at the final follow-up post-surgery, their scores improved, with two patients achieving an excellent score of 2.44 and one achieving a good score of 4.08 (Fig. 3a). All patients showed improved Quick Disability of the Arm, Shoulder, and Hand scores, which represented hand function recovery (mean, 78 pre-surgery; mean, 30.3 at final follow-up) (Fig. 3b). All patients had poor static two-point discrimination in the injured nerve distribution (20 mm) pre-surgery, which improved to better than good (10, 3, and 4 mm) at the final follow-up (Fig. 3c). All patients had poor pre-surgery moving m2PD in the injured nerve distribution (20 mm). One patient still had poor m2PD and two improved to excellent m2PD (10, 2, and 3 mm) at the final follow-up (Fig. 3d).

Patients with peripheral nerve trauma are often young and active; therefore, subsequent loss of function is particularly devastating[22]. Therefore, any treatment promoting functional recovery after peripheral nerve trauma is of great value[22]. The nerve segment is lost in some injuries, and the nerve stump cannot be directly sutured without tension. In these cases, an autologous nerve graft bridges the gap[22]. However, autologous nerve grafts require healthy nerve harvesting, which may lead to donor site morbidity, such as donor nerve sensation loss, neuroma formation, and pain[1,2]. Artificial nerve conduits have been developed to avoid these conditions[23,24]. However, the nerve regeneration potential of artificial nerve conduits is not as high as that of autologous nerve grafts due to the lack of cellular elements. From the tissue engineering perspective, nerve tissue regeneration requires scaffolds, cells, and growth factors[24]. Artificial nerve conduits contain only an extracellular matrix (ECM) without cellular elements. For better nerve regeneration, many researchers have added Schwann cells (SCs) or mesenchymal stem cells to artificial conduits and reported some improvement in nerve regeneration[1,2]. However, the death of the injected cells and leakage from the artificial conduit limit the viability and seeding efficacy of supportive cells[25]. Therefore, a biological, tissue-engineered, and scaffold-free conduit (Bio 3D conduit) was developed previously, and its usefulness for nerve regeneration in rat and canine models was reported[1–7]. Based on these findings, we initiated the current investigator-initiated clinical trial. The patients had no adverse events, which was the primary study outcome, and achieved nerve recovery, the secondary outcome.

SCs and fibroblasts are essential for axonal regrowth[26]. Parrinello found that after nerve injury, coordinated ephrin-B/EphB2 signalling between SCs, and fibroblasts results in directional cell

**Table 1 Patient's demographic data, site of the nerve lacerations, and defect length.**

| Case | Age (years) | Sex | Injury mechanism | Location | Injured nerve | Duration of injury (months) | Associate injuries | Nerve defect (mm) |
|---|---|---|---|---|---|---|---|---|
| B3CON01 | 46 | M | Galvanised sheet | Left wrist | Superficial branch of ulnar nerve | 7 | Median nerve Flexor tendon | 5 |
| B3CON02 | 50 | M | Hand saw | Left ring finger | Digital nerve | 5 | Digital artery | 1 |
| B3CON03 | 34 | M | Table saw | Left dorsal hand | Superficial branch of radial nerve | 1 | Open bone fracture Extensor tendon | 1 |

**Table 2 Adverse event occurrence.**

| System organ class | Preferred terms | Grade | Number of occurrences | Rate of occurrences (%) |
|---|---|---|---|---|
| Gastrointestinal disorders | Constipation | 1 | 2 | 66 |
| | Nausea | 1 | 1 | 33 |
| | Vomiting | 1 | 1 | 33 |
| General disorders and administration site conditions | Pain | 1 | 1 | 33 |
| Injury, poisoning, and procedural complications | Application site pain | 1 | 1 | 33 |
| | Wound complication | 1 | 3 | 100 |

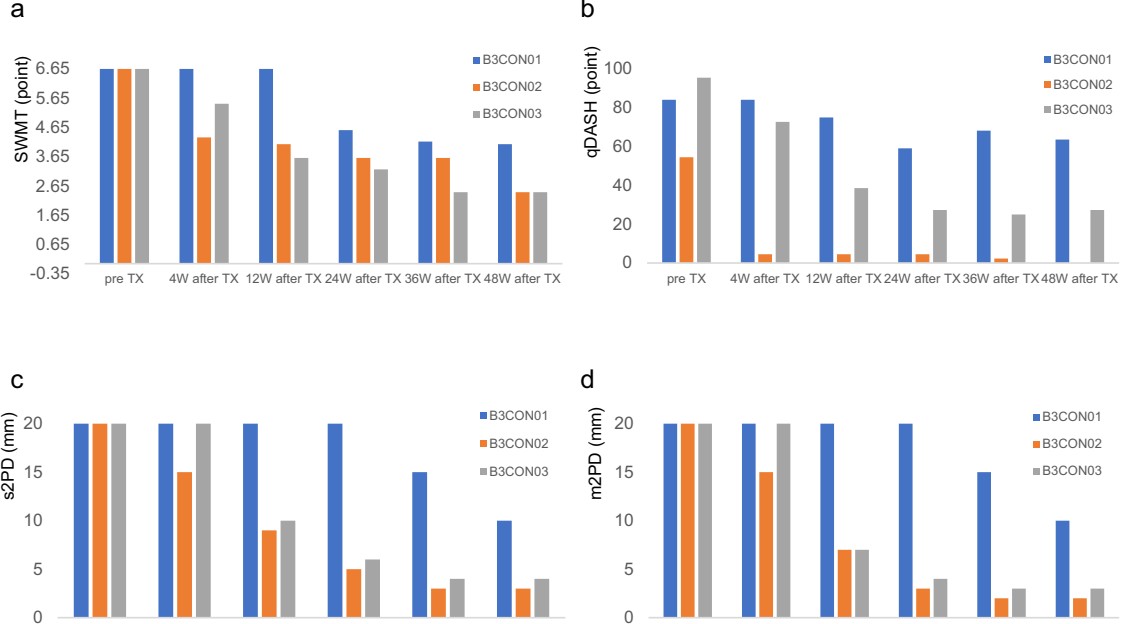

**Fig. 3 Patients' clinical trial results. a** Results of the Semmes–Wein Monofilament test (SWMT). All patients had poor SWMT score (6.65) before surgery, which improved to more than good SWMT score (4.08, 2.44, and 2.44) at the final follow-up. **b** Quick Disability of the Arm, Shoulder, and Hand (qDASH) score The qDASH score improved in all patients, indicating functional hand recovery. **c** Results of the static two-point discrimination (s2PD) in the injured nerve distribution. All patients had poor s2PD (20 mm) before surgery, which improved to more than good s2PD (10, 3, and 4 mm) at the final follow-up. **d** Improvement in moving two-point discrimination improvement (m2PD) in the injured nerve distribution. All patients had poor m2PD (20 mm) before surgery, and two patients improved to excellent m2PD (2 and 3 mm) at the final follow-up. The numerical data presented here can also be found in Supplementary Data 1.

migration of SCs out of nerve stumps to guide regenerating axons[26]. The human dermal fibroblasts differentiated into functional SCs or mesenchymal stem cells[27,28]. Moreover, differentiation of dermal fibroblasts into SC-like cells in regenerated nerves was reported[3]. Additionally, fibroblasts produce an ECM that provides a well-defined physical and chemical environment that promotes cell survival, differentiation, and homeostasis[29]. The use of ECM-modified scaffolds is a good approach for peripheral nerve repair as it improves endogenous cell survival and migration to the injury site[30]. We selected fibroblasts to create the Bio 3D nerve conduits based on these data.

In conclusion, we inducted Bio 3D nerve conduits in an investigator-initiated clinical trial. The patients did not experience any adverse events caused by the Bio 3D nerve conduits. All patients exhibited nerve recovery. Bio 3D nerve conduits may be a viable treatment option for peripheral nerve injuries.

## Data availability
The clinical results of this study are shown in Fig. 1, Supplementary Data 1 and Tables 1 and 2. The remaining data dataset generated during the current study are available from the corresponding author upon reasonable request, after approval by the data access committee. This approval is required because the patients' data is confidential.

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

## Acknowledgements

The authors thank the staff involved in this trial at CCMT and iACT at Kyoto University. This study was partly funded by the Japan Agency for Medical Research and Development (AMED) under grant number 21lm0203121h0002.

## Author contributions

R.I., T.N., T.A., M.U., Y.A., A.N., N.M., S.Y., M.N.G., K.M., Y.A., Y.T., Y.M., S.A., and S.M. contributed to important intellectual content. R.I., T.A., M.U., and S.A. conceived and designed the study. T.A., K.M., Y.A., Y.T., Y.M., and Y.A. performed the experiments. R.I., T.N., N.M., and S.Y. were involved in clinical management. A.N. performed the data analysis. S.M. was responsible for conducting the clinical trial. M.N., R.I., T.A., Y.A., R.I., T.A., and M.N.G. drafted the manuscript. All the authors have written, reviewed, and approved the manuscript. All the authors critically reviewed the manuscript and contributed important intellectual content.

## Competing interests

The authors declare the following competing interests: S.A., Y.M., and Y.T. are employees of Cyfuse Biomedical KK, who contributed to the fabrication of the 3D conduits by providing the bioprinter. The other authors declare no competing interests.
