## [Peer Review File · Communications Medicine]

Reviewers' comments:

Reviewer #1 (Remarks to the Author):

The authors describe their efforts to develop a scaffold free Bio 3D conduit and the results from early human clinical trials. I found this manuscript to be well done and very interesting, although I would say that the title claiming to be scaffold free seems to be misleading as a silicone tube was used in the perfusion bioreactor as a form of a scaffold - do the authors think the title accurately reflects the process? Additionally, I have some specific comments below:

Line 14: This is awkward, please reword "Previous studies have reported the creation o Bio 3D nerve conduits using a"

Line 40: Where was the skin harvested from? Describe the isolation and expansion of the dermal fibroblasts or provide a reference. You need to also explain how the Bio 3D nerve conduits were made on the printer. What parameters, what size? Where they tailored to the individual defect? I assume each patient had their own fibroblasts used? Please specify this.

Line 41 – Do you mean intraoperatively you exposed the nerves?

Line 54: Did you do 2 point in the distribution of the nerves repaired or was this the average digital 2 point? This matters because many of those nerves do not contribute to typical 2 PD measurements.

Line 81: Did you expand the dermal fibroblasts to remain fibroblasts or did you transform these into Schwann cells?

Line 87: Did you ever measure the mechanical properties of these conduits? If so, it would be appropriate to discuss this here.

Also, how did you sterilize the scaffold after fabrication?

Figure 2 A: I believe you have a typo – "Medicant nerve"

Figure 2 C-F: Please label the y-axes on the graphs to make them easier to understand.

Motor nerve function analysis – None of these assessments would be affected by the nerve injury and therefore are not relevant to this study.

Reviewer #2 (Remarks to the Author):

This manuscript describes the safety and feasibility of the treatment of a nerve injury in humans with a fibroblast-populated nerve conduit.

The development of a living nerve conduit using autologous fibroblasts is of interest, and the safety of its transplantation to humans is relevant to study.

However, the challenge to repair a small 2cm nerve gap of a small ulnar nerve is limited. Such repair could be performed with already commercialized collagen nerve tubes. It would have been informative and relevant to compare results already published with these nerve tubes grafted in

humans with the present results.

Indeed, one major limitation of the fabrication of this living nerve tube is the cost of the autologous fibroblast purification and culture process, in addition to the length of the preparation, 60 days, which seem to be a major drawback compared to collagen tubes. Moreover, such manipulations may increase the risk of contamination compared to sterilized tubes.

Thus, a major improvement of motor and sensory functions recoveries should be shown to justify the use of this construct in humans, such as a beneficial effect on larger and longer nerve gaps.

Why the diameter of the conduit has a so high variability: $2.5 \pm 1\text{mm}$?

About the sentence "The perfusion cultivation was continued until the desired function and strength were achieved.", it was not clear how these desired function and strength were assessed.

The size of the skin biopsy harvested seems high and may induce scarring. A smaller 6mm round punch biopsy is usually sufficient for fibroblast purification and banking.

Figures C-F lack information about what is analyzed in the Y-axis. Pictures and legends of figure 1C are not enough informative.

We are very grateful to the editors for providing such valuable, thoughtful, and thorough comments. The comments and questions have prompted us to re-evaluate our data from new and interesting angles, which has considerably strengthened the work. Thank you.

Reviewers' comments:

Reviewer #1 (Remarks to the Author):

[Comment #1]

The authors describe their efforts to develop a scaffold free Bio 3D conduit and the results from early human clinical trials. I found this manuscript to be well done and very interesting, although I would say that the title claiming to be scaffold free seems to be misleading as a silicone tube was used in the perfusion bioreactor as a form of a scaffold - do the authors think the title accurately reflects the process? Additionally, I have some specific comments below:

[Author action #1]

We apologize for the inadequate explanation. Although cells may have come in contact with the silicone tubes of the bioreactor during processing, the final product did not contain the artificial material scaffold. We believe that the three-dimensional nerve conduit used in current study is unique in that it is constructed entirely of cells without any artificial material. We propose that the title remain unchanged.

[Comment #2]

Line 14: This is awkward, please reword "Previous studies have reported the creation o Bio 3D nerve conduits using a"

[Author action #2]

Thank you for pointing this out. Accordingly, we have rewritten the corresponding sentence.

[Comment #3]

Line 40: Where was the skin harvested from? Describe the isolation and expansion of the dermal fibroblasts or provide a reference. You need to also explain how the Bio 3D nerve conduits were made on the printer. What parameters, what size? Where they tailored to the individual defect? I assume each patient had their own fibroblasts used? Please specify this.

[Author action #3]

We apologize for the unclear explanation. The description of skin harvesting to preparation of Bio 3D nerve conduits is described in detail in the methods section.

[Comment #4]

Line 41 – Do you mean intraoperatively you exposed the nerves?

[Author action #4]

Yes, the nerves were exposed intraoperatively. This has also been described in detail in the methods section.

[Comment #5]

Line 54: Did you do 2 point in the distribution of the nerves repaired or was this the average digital 2 point? This matters because many of those nerves do not contribute to typical 2 PD measurements.

[Author action #5]

Thank you for your valuable questions. Two-point discrimination was performed in the repaired nerves. According to your suggestion, we have added the words “in the distribution of the injured nerve”.

[Comment #6]

Line 81: Did you expand the dermal fibroblasts to remain fibroblasts or did you transform these into Schwann cells?

[Author action #6]

Thank you for your pertinent comment. We expanded the dermal fibroblasts but not to transform them into Schwann cells. This is also described in detail in the methods section.

[Comment #7]

Line 87: Did you ever measure the mechanical properties of these conduits? If so, it would be appropriate to discuss this here.

[Author action #7]

Thank you for your valuable remark. The mechanical property of the conduit was determined by cutting out a portion of the sample and performing a suture test. We mentioned this in the Method section.

[Comment #8]

Also, how did you sterilize the scaffold after fabrication?

[Author action #8]

This conduit was composed entirely of cells; therefore, scaffolds were not used. The tubes of the bioreactor were sterilized and new ones were used for each preparation.

[Comment #9]

Figure 2 A: I believe you have a typo – “Medicant nerve”

[Author action #9]

I apologize for the error. We corrected the typographical error as “Median nerve”.

[Comment #10]

Figure 2 C-F: Please label the y-axes on the graphs to make them easier to understand.

[Author action #10]

We apologize for missing out on this detail. Accordingly, we have added explanatory text for the Y-axis in Fig. 2 C-F.

[Comment #11]

Motor nerve function analysis – None of these assessments would be affected by the nerve injury and therefore are not relevant to this study.

[Author action #11]

We agree that a motor function test was not necessary in this case. However, we set this as a reference endpoint because we anticipated that some patients with motor nerve damage could enter this trial.

Reviewer #2 (Remarks to the Author):

[Comment #1]

This manuscript describes the safety and feasibility of the treatment of a nerve injury in humans with a fibroblast-populated nerve conduit.

The development of a living nerve conduit using autologous fibroblasts is of interest, and the safety of its transplantation to humans is relevant to study.

However, the challenge to repair a small 2cm nerve gap of a small ulnar nerve is limited. Such repair could be performed with already commercialized collagen nerve tubes. It would have been informative and relevant to compare results already published with these nerve tubes grafted in humans with the present results.

Indeed, one major limitation of the fabrication of this living nerve tube is the cost of the autologous fibroblast purification and culture process, in addition to the length of the preparation, 60 days, which seem to be a major drawback compared to collagen tubes. Moreover, such manipulations may increase the risk of contamination compared to sterilized tubes.

Thus, a major improvement of motor and sensory functions recoveries should be shown to justify the use of this construct in humans, such as a beneficial effect on larger and longer nerve gaps.

[Author action #1]

Thank you for your valuable comments. As you pointed out, collagen tubes for cross-linking 2 cm nerve defects are commercially available, but they have not led to favorable clinical results. Although the 60-day preparation period and the lack of sterilization are the associated disadvantages, the nerve regeneration capacity of this product, which is comparable to that of autologous nerve grafts, is higher than that of collagen tubes. Since this is the first in-human study, we conducted the trial on a 2 cm nerve defect to first confirm the safety and efficacy.

[Comment #2]

Why the diameter of the conduit has a so high variability: 2.5 ± 1 mm?

[Author action #2]

The current trial was designed to assess the efficiency of the repair of peripheral nerve damage with a diameter of 1.5 to 2 mm. Therefore, the diameter of conduit was defined as 2.5 ± 1 mm.

[Comment #3]

About the sentence "The perfusion cultivation was continued until the desired function and

strength were achieved.”, it was not clear how these desired function and strength were assessed.

[Author action #3]

Thank you for your valuable remark. The mechanical property of the conduit was determined by cutting out a portion of the sample and performing a suture test. We mentioned this in the Method section.

[Comment #4]

The size of the skin biopsy harvested seems high and may induce scarring. A smaller 6mm round punch biopsy is usually sufficient for fibroblast purification and banking.

[Author action #4]

Thank you for your pertinent comment. We agree that skin sampling is a burden to the patient. However, in case of autologous cultured skin grafts (attached Guide for autologous cultured skin graft), the average size of the biopsy is 1 cm². Based on this, we do not believe that the size of the skin biopsy in this case was large. We also believe that the wound heals better when the biopsy is performed in a boat shape and sutured, compared to that in a punch shape.

[Comment #5]

Figures C-F lack information about what is analyzed in the Y-axis. Pictures and legends of figure 1C are not enough informative.

[Author action #5]

We apologize the lack of Y-axis in Fig. 2 C-F. Accordingly, we have added explanatory text for the Y-axis in the figure.

Further, we apologize for the inadequate description in the figure legend; we have added subsections a to h in Fig. 1C. Additionally, length bars have been added to Fig. 1 C-b and Fig. 1 C-c.

REVIEWERS' COMMENTS:

Reviewer #1 (Remarks to the Author):

The authors appear to have satisfactorily responded to the concerns raised.